# Plant Proteins as Alternative Natural Emulsifiers in Food Emulsions

**DOI:** 10.3390/foods14244291

**Published:** 2025-12-13

**Authors:** Dominika Kaczmarek, Marta Pokora-Carzynska, Leslaw Juszczak, Ewelina Jamroz, Janusz Kapusniak

**Affiliations:** 1Department of Dietetics and Food Studies, Faculty of Science and Technology, Jan Dlugosz University in Czestochowa, Armii Krajowej 13/15, 42-200 Czestochowa, Poland; dominika.kaczmarek@doktorant.ujd.edu.pl (D.K.); m.pokora-carzynska@ujd.edu.pl (M.P.-C.); l.juszczak@ujd.edu.pl (L.J.); 2Department of Chemistry, Faculty of Food Technology, University of Agriculture in Krakow, Balicka 122, 30-149 Krakow, Poland; ewelina.jamroz@urk.edu.pl; 3Department of Packaging and Logistics Processes, Krakow University of Economics, Rakowicka 27, 31-510 Krakow, Poland

**Keywords:** rice proteins, pea proteins, chickpea proteins, emulsion stabilization, physicochemical properties, foaming properties, emulsification, water and oil absorption capacity, storage stability

## Abstract

The growing interest in plant-based ingredients in food production has increased the demand for effective alternatives to animal-derived emulsifiers. In this study, the physicochemical and functional properties of selected commercial plant protein preparations as natural emulsifiers in food emulsions were assessed. Emulsifying activity and stability (EA, ES), foaming capacity and stability (FC, FS), water and oil absorption (WAC, OAC), color (CIE *Lab**), viscosity, surface tension, and zeta potential were analyzed. Pea (PP1–PP4), rice (RP1, RP2) and chickpea (CP1) proteins showed the most favorable properties, characterized by high EA values (58.3–62.5%) and emulsion stability during storage (62–65%) after 6 days. Emulsions formulated with these proteins were significantly lighter (*L** > 69). PP1 exhibited more than twice the viscosity of the other samples. The lowest surface tension values (<45 mN/m) were observed for RP2 and PP1, indicating strong surface activity. Pea proteins PP1, PP2, and PP4 showed the highest system stability, with zeta potential values below –35 mV. Overall, the selected plant protein preparations, particularly pea, rice, and chickpea proteins, showed promising functional properties, confirming their potential use as natural emulsifiers in clean-label plant-based formulations and providing a basis for further product development.

## 1. Introduction

The modern food industry faces a growing demand for natural, sustainable, and functional ingredients that can replace synthetic additives and animal-derived ingredients [1]. Emulsifiers play a key role in stabilizing oil–water emulsions by adsorbing at the interface, lowering surface tension, and forming a protective layer around fat droplets, preventing them from merging. In products such as sauces, instant beverages, creamers, and fermented dairy products, emulsifiers are responsible for uniform consistency, creaminess, and extended shelf life [2,3].

Commonly used emulsifiers, such as lecithin, milk proteins (e.g., casein, whey), and monoglycerides and diglycerides of fatty acids, effectively stabilize emulsions, but they are increasingly controversial among consumers and are unacceptable in a vegan diet or in the context of food allergies [4,5]. In response to the needs of vegans, allergy sufferers, and consumers seeking a “clean label,” the food industry is increasingly using plant proteins, polysaccharides, microalgae emulsifiers, and innovative Pickering systems as effective and safe substitutes for traditional animal and synthetic emulsifiers [6,7,8,9].

Plant proteins such as pea, soy, faba bean, and bean proteins have the ability to stabilize oil–water emulsions due to their amphiphilic nature, but their emulsifying activity and stability are typically lower than those of dairy proteins (e.g., casein, whey). This is due to differences in structure, solubility, molecular weight, and adsorption capacity at the interface [10,11].

Despite these limitations, the literature indicates a growing interest in using plant proteins as natural emulsifiers. The use of appropriate processing methods, such as physical (e.g., sonication, high pressure), chemical (e.g., glycosylation, acetylation), and enzymatic modifications, can significantly improve the solubility, surface properties, emulsifying capacity, and bioavailability of plant proteins, bringing their functionality closer to that of animal proteins [12,13,14]. Enzymatic hydrolysis is a particularly promising technique, which, as recent research by Bekiroglu et al. [15] indicates, can lead to controlled modification of protein structure, increasing their affinity for the oil phase, improving surface activity, and stabilizing the oil–water interface. The aim of this study was to evaluate the physicochemical and functional properties of eleven commercial plant protein preparations, derived from pea, rice, chickpea, hemp, sunflower, and oat, used in their unmodified form, as potential emulsifiers in the production of plant-based foods. The obtained results allow for the identification of protein preparations with the most beneficial properties and will form the basis for further research, including the design of stable plant emulsions. In the future, these emulsions may be used in processing methods, such as spray drying, aimed at developing alternative instant coffee whiteners.

Compared to previous research, this work presents a comprehensive assessment of the physicochemical and functional properties of selected plant proteins, with particular emphasis on their technological suitability for creating stable emulsions. Analysis encompassing such a broad range of parameters allows for a better understanding of the differences between plant-based raw materials and their application potential in various technological processes.

The results collected can provide a valuable starting point for further research, including the design of model products based on plant emulsions. Future work also plans to expand the analysis to include an assessment of the behavior of these systems under specific processing conditions, enabling a more comprehensive characterization of their application potential.

## 2. Materials and Methods

### 2.1. Materials

The study material consisted of eleven commercial plant-based protein preparations in powdered form (isolates or concentrates) from various manufacturers. The study group included rice, pea, hemp, chickpea, sunflower, and oat proteins. All protein preparations were commercially available from the producers in the year of their acquisition (2023). The collected research material consisted of samples purchased directly from producers or distributors via online stores, as well as those obtained free of charge for research purposes. Detailed information on the producers, geographic locations, declared protein content, and classification of the raw material as an isolate or concentrate is presented in Table 1.

It is worth emphasizing that all protein preparations used were analyzed in their unaltered form, without any prior technological modifications, allowing for a reliable assessment of their natural physicochemical and functional properties.

### 2.2. Methods

#### 2.2.1. Determination of Water Absorption Capacity (WAC)

A 1.00 ± 0.01 g sample of the protein preparation was thoroughly mixed with 30 mL of distilled water using a microshaker and then homogenized at 1000 rpm using a homogenizer (IKA, Staufen, Germany). The sample was then centrifuged in a centrifuge (MPW-380, MPW Med. Instruments, Warsaw, Poland) at 9000× *g* for 15 min. After separation of free water, the tubes with the sediment were placed upside down on filter paper for 10 min to allow the residual water to drain [16,17]. The water absorption capacity was calculated according to the following equation:
(1)$$\mathrm{WAC}=\frac{a}{W}$$ where WAC—water absorption capacity [g H_2_O/g sample], *a*—wet sediment mass [g], *W*—weight of protein preparation [g].

Results are expressed as grams of water absorbed per gram of protein preparation. All determinations were performed in triplicate.

#### 2.2.2. Determination of Oil Absorption Capacity (OAC) and Water–Oil Absorption Index (WOAI)

Oil absorption capacity (OAC) was determined according to the procedure described by Lin et al. [18]. 5.00 ± 0.01 g of protein preparation was weighed into a Falcon tube, and 25 mL of refined rapeseed oil (Fat Plants, Szamotuly, Poland) was added. The mixture was mixed using a microshaker and then homogenized in a manner analogous to that used for the water absorption method. The sample was then allowed to stand for 5 min at room temperature (~22 °C) and then rehomogenized (1 min). After the second mixing, the samples were centrifuged in a centrifuge (MPW-380, MPW Med. Instruments, Warsaw, Poland) at 9000× *g* for 15 min. The unbound oil was carefully poured into a graduated cylinder, and the volume of decanted oil (b) was recorded in mL. The fat absorption capacity was calculated using the formula:
(2)$$\mathrm{OAC}=\frac{25-b}{W}$$ where OAC—oil absorption capacity [mL of oil/g of sample], 25—number of mL of oil used for the determination, *b*—volume of decanted oil [mL], *W*—weight of protein preparation [g].

Results were expressed as the amount of oil [mL] absorbed by 1 g of protein preparation. All determinations were performed in triplicate.

The Water–Oil Absorption Index (WOAI) was calculated as the ratio of water absorption capacity (WAC) to oil absorption capacity (OAC).

#### 2.2.3. Determination of Foaming Capacity (FC) and Foam Stability (FS)

Foaming capacity (FC) and foam stability (FS) were determined according to a modified method described by Zhang et al. [19]. A mass of 1.00 ± 0.01 g of the protein preparation was weighed into a beaker, and 99 mL of distilled water at room temperature (~22 °C) was added. The mixture was homogenized (IKA, Staufen, Germany) for 1 min at 10,000 rpm. Immediately after homogenization, the foamed liquid was carefully transferred to graduated cylinder, and the total volume (liquid + foam) was read. Foaming capacity (FC) was calculated as the percentage increase in volume after homogenization compared to the initial volume (100 mL):
(3)$$FC=\frac{V_{1}-\;V_{0}}{V_{0}}\;\cdot 100$$ where FC—foaming capacity [%], *V*_0_—initial volume (100 mL), *V*_1_—volume after foaming (liquid + foam) [mL].

After 10 min, the remaining foam volume (*V*_2_) was read, and the foam stability (FS) was calculated according to the formula:
(4)$$FS=\frac{V_{2}-\;V_{1}}{V_{0}}\;\cdot 100$$ where FS—foam stability [%], *V*_2_—volume after foaming after 10 min [mL].

All determinations were performed in triplicate.

#### 2.2.4. Determination of Emulsification Activity (EA) and Emulsion Stability (ES)

Emulsifying activity (EA) and emulsion stability (ES) were determined using the method described by Yasumatsu et al. [20] with modifications. To 15 mL of distilled water was added 0.75 g of the protein preparation under study, followed by 15 mL of refined rapeseed oil (Fat Plants, Szamotuly, Poland). The mixture was homogenized for 1 min at 5000 rpm. The resulting emulsion was evenly distributed into centrifuge tubes and centrifuged at 3897× *g* for 5 min using a centrifuge (MPW-380, MPW Med. Instruments, Warsaw, Poland). After completion of the process, the volume of the stable emulsion phase (V_E1_) and the total sample volume (V_T_) were read. Emulsification activity (EA) was calculated according to the equation [21]:
(5)$$EA=\frac{V_{E1}}{V_{T}}\cdot 100$$ where EA—emulsification activity [%], *V_E_*_1_—volume of the stable emulsion phase [mL], *V_T_*—total sample volume [mL].

To determine emulsion stability (ES), the prepared emulsion was heated in a water bath for 30 min at 30 °C before centrifugation, then cooled to room temperature and centrifuged under the same conditions. ES was calculated according to the formula [22]:
(6)$$ES=\frac{V_{E2}}{V_{T}}\cdot 100$$ where ES—emulsion stability [%], *V_E_*_2_—volume of the stable emulsion phase after 30 min of heating at 30 °C [mL], *V_T_*—total sample volume before heat treatment [mL].

All determinations were performed in parallel in triplicate.

#### 2.2.5. Determination of Storage Stability of Emulsions (SP)

Emulsion samples were stored for 6 days under refrigerated conditions (4 ± 1 °C). Measurements were taken after 3 h and after 2, 4, and 6 days of storage. The storage stability of the emulsions was expressed numerically using a method based on phase separation measurement:
(7)$$SP=\frac{V_{E3}}{V_{T}}\cdot 100$$ where SP—storage stability [%], *V_E_*_3_—volume of the stable emulsion phase after a specified storage time [mL], *V_T_*—total sample volume [mL].

All measurements were performed in triplicate.

#### 2.2.6. Color Determination of Plant Emulsions Using the CIE *L*a*b** Model

The color of the prepared plant emulsions (as per Section 2.2.5) was determined using a colorimeter (Chroma Meter CR-400, Konica Minolta Sensing, Osaka, Japan) relative to the standard illuminant D65 and 10°, using the three-dimensional CIE *L*a*b** color model. Three measurements were performed for each emulsion sample.

In addition, the total color difference (Δ*E*) between each emulsion and the reference white standard plate was calculated according to the CIE 1976 formula:
(8)$$\Delta E=\sqrt{(L^{*}-\;L_{ref}^{*}{)}^{2\;}+(a^{*}-\;a_{ref}^{*}{)}^{2\;}+\;(b^{*}-\;b_{ref}^{*}{)}^{2\;}}\;$$ where
$L_{ref}^{*}$—color brightness,
$a_{ref}^{*}$—chromaticity in the red–green range,
$b_{ref}^{*}$—chromaticity in the yellow–blue range.

#### 2.2.7. Determination of the Viscosity of Protein Solutions

The dynamic viscosity of protein solutions at a concentration of 5 g/100 mL (*w*/*v*) at 25 ± 1 °C was determined using an Ubbelohde capillary viscometer (53203/OC, SI Analytics GmbH, Mainz, Germany). Flow time was measured using an automatic ViscoClock plus meter (SI Analytics GmbH, Mainz, Germany). Based on the determined flow times and the viscometer constant, kinematic viscosity values were calculated and then converted to dynamic viscosity. The density of the tested solutions was measured at 25 ± 1 °C using a densimeter (Densito 30PX, Mettler Toledo, Greifensee, Switzerland). The measurements were performed in triplicate.

#### 2.2.8. Surface Tension Determination

The surface tension of protein solutions at a concentration of 5 g/100 mL (*w*/*v*) at room temperature (~22 °C) was determined by the suspended drop method using an OCA 25 goniometer equipped with SCA20 software ver. 5.0.41 (DataPhysics Instruments GmbH, Filderstadt, Germany). The measurements were performed in ten replicates.

#### 2.2.9. Zeta Potential Determination

The zeta potential of 100 mg/100 mL (*w*/*v*) protein solutions at room temperature (~22 °C) was determined by dynamic light scattering (DLS) (Zetasizer Ultra Red, Malvern Instruments Ltd., Malvern, UK). Measurements were performed in triplicate, following the procedure described by Jamróz et al. [23].

#### 2.2.10. Statistical Analysis

The Statistica v.13.0 software was used for data analysis. Mean values and standard deviations were calculated. The data were subjected to variance analysis (ANOVA). Duncan’s multiple-range test was used for mean comparison. The *p* < 0.05 level was considered statistically significant. Additionally, Pearson’s linear correlation coefficients were calculated between selected parameters characterizing the analyzed protein preparations, and their statistical significance was assessed at 0.05 level.

## 3. Results and Discussion

### 3.1. Determination of Water Absorption Capacity (WAC) and Oil Absorption Capacity (OAC)

Protein sorption properties, such as water absorption capacity (WAC) and oil absorption capacity (OAC), are of significant technological importance, particularly in the context of formulating emulsions and instant products. These parameters influence, among other things, the texture, stability, and functionality of finished food systems [24]. The results of the analyzed protein preparations are summarized in Table 2.

Among the analyzed preparations, the highest water-binding capacity was demonstrated by pea proteins: PP1 (6.81 ± 1.03 g H_2_O/g protein), PP4 (5.50 ± 0.76 g H_2_O/g protein), PP3 (4.66 ± 0.16 g H_2_O/g protein), and PP2 (3.97 ± 0.01 g H_2_O/g protein). Similar results were obtained in the studies by Schumacher et al. [25], which may confirm the beneficial sorption properties of pea proteins. Sunflower, oat, chickpea, and rice proteins also demonstrated relatively good water absorption capacity. For these preparations, absorption values ranged from 2.97 to 3.87 g H_2_O/g protein. The obtained results indicate that preparations of these proteins may be useful as structural components in applications where moisture retention is important, e.g., in instant products or dairy analogs [26,27,28,29]. In turn, the lowest water absorption capacity was demonstrated by hemp proteins HP1 and HP2 (WAC 2.44 g H_2_O/g protein). This may be due to their higher content of insoluble fractions and dietary fiber, which reduces the effective contact surface with water and limits the possibility of hydration [30,31]. Water absorption capacity significantly correlated with foaming capacity (r = 0.6753, *p* < 0.05), indicating that proteins with higher hydration ability generally formed foam more effectively.

Hemp proteins also proved to be the least effective in fat absorption (OAC 1.70–1.90 mL oil/g protein). Oil absorption capacity (OAC) is one of the key functional properties of proteins used in food technology, particularly in the context of oil–water emulsions. This parameter indicates the preparation’s ability to bind lipids and form stable fat structures, directly influencing sensory properties such as creaminess, palatability, and texture of finished products [24]. In the conducted studies, the highest oil absorption capacity (OAC) was observed for pea protein PP3 (2.40 ± 0.01 mL oil/g protein) and chickpea protein CP1 (2.45 ± 0.07 mL oil/g protein). This property results from the presence of surface-active, hydrophobic protein fractions. These fractions promote adsorption at the oil–water interface, which translates into high affinity for lipids and favorable emulsifying properties [32,33]. Rice proteins also demonstrated relatively high fat absorption (RP1 and RP2—2.10 mL oil/g protein each).

Oil absorption capacity correlated significantly (*p* < 0.05) and positively with FC (r = 0.6995), FS (r = 0.7171), EA (r = 0.7169), and ES (r = 0.7312), confirming that proteins with greater affinity for lipids also displayed better foaming and emulsifying properties.

Additionally, OAC showed a significant positive correlation with lightness (*L**) (*p* < 0.05), suggesting that lighter protein preparations exhibited a higher capacity for oil absorption. High OAC values may support the formation of smooth and rich textures, particularly desirable in instant products such as vegan coffee creamers.

The WAC-to-OAC ratio predicts a protein’s ability to stabilize an emulsion. According to literature, the optimal WOAI should be approximately 2:1, meaning that the protein should bind twice as much water as fat, achieving a good balance between hydrophilic and lipophilic surface properties. Excessive hydrophilicity or lipophilicity reduces emulsification ability [34]. Most of the protein preparations analyzed met this requirement.

When used in instant products such as creamers, a high water absorption capacity (WAC) is particularly desirable, as it ensures rapid hydration after pouring and appropriate texture. At the same time, a moderate OAC helps maintain the product’s creaminess and smoothness without excessive fat separation.

### 3.2. Foaming Properties

The foaming properties of the tested proteins are crucial, for example, in products such as vegan coffee creamers/frothers, because they determine the texture and stability of the foam and thus the sensory quality of the drink. Foam formation is a process dependent on three factors: the transport of molecules to the air–water interface, their adsorption and penetration, and the reorganization of molecules at this interface [19]. The ability to form foam and its stability varied depending on the type of plant protein preparation used (Figure 1). In general, FC correlated positively with FS, indicating that higher foaming capacity was associated with improved foam stability.

The highest foaming capacity (FC) was achieved for emulsions containing pea proteins (PP1 and PP3), with FC values of 70% and 69%, respectively. The high surface activity of these proteins favored a rapid reduction in the interfacial tension at the air–water interface, facilitating foam formation. The high foaming capacity of these proteins may result from their favorable surface-active properties and the appropriate degree of denaturation, which favors a rapid reduction in the interfacial tension at the air–water interface. Similar observations regarding the high foaming capacity of pea protein were made by Stamatie et al. [35] and Shevkani et al. [36], who found that pea proteins are characterized by good structural flexibility, facilitating the formation of stable interfacial films. Additionally, the foaming properties are influenced by the elasticity and rheology of the interfacial film, which stabilizes air bubbles. Proteins with a greater ability to partially unfold their structures form more elastic and deformation-resistant films, which reduces foam coalescence and drainage. Proteins that partially unfold at the interface form more flexible and deformation-resistant films, increasing foam stability by preventing foam rupture and bubble fusion [37].

Foaming capacity showed significant positive correlations with emulsifying activity (EA) (r = 0.7592) and emulsifying stability (ES) (r = 0.7720). This relationship is expected, as the mechanisms underlying air-bubble stabilization in foams are analogous to those responsible for fat-droplet stabilization in emulsions.

Plant albumins (e.g., pea albumins) exhibit excellent foaming properties, often comparable to or superior to those of whey or egg proteins. Globulins, on the other hand, produce foams of low stability, primarily due to their tendency to aggregate and weaker interfacial layer formation [38].

High values were also recorded for the remaining pea proteins (PP2 and PP4) and rice protein (RP2), which had foaming capacity of 55–56%. This is confirmed by the research by Cho et al. [39], who showed that rice bran proteins are superior to soy proteins in terms of foam formation and stability. In contrast, chickpea protein (CP1) was characterized by moderate foaming capacity (37%) and medium foam stability (25%), which may be attributed to the presence of fractions with limited solubility and a greater tendency to form rigid protein structures at the interface.

The lowest foaming capacity and foam stability were observed for hemp (HP1, HP2), sunflower (SP1), and oat (OP1) proteins, with FC values not exceeding 10%. The poor foam-forming ability of these proteins may be due to limited flexibility of the protein structures, which hinders the formation of stable interfacial films [40,41,42]. Furthermore, these preparations are characterized by a relatively high fat content, which acts as a foaming inhibitor. The presence of lipids at the interface can compete with proteins for adsorption sites or lead to destabilization of forming films by disrupting their continuity and reducing their mechanical resistance. Consequently, both the formation of effective foam and its maintenance over time are significantly hindered [43,44].

Foam stability (FS) was highest in samples containing pea protein (PP3 and PP4) and rice protein, confirming that these raw materials are characterized by a good ability to stabilize foam by forming flexible protein films with gelling properties [45,46]. The exception was sample PP1, which, despite the highest foaming capacity (FC), showed the largest decrease in foam volume after 10 min, indicating low foam stability.

FS also correlated positively with EA (r = 0.6592) and ES (r = 0.6717), supporting the interpretation that interfacial phenomena governing emulsion and foam stability share common mechanistic foundations. A high FS value shortly after whipping is particularly desirable in the context of developing products such as vegan coffee creamers, where a stable, light foam that persists during consumption is desired [47]. The obtained results indicate that pea and rice proteins have the greatest application potential in this type of product, which results from their favorable surface properties and appropriate hydrophobic–hydrophilic balance, enabling foam stabilization within a short time after product preparation.

### 3.3. Emulsifying Properties

Protein emulsification is the ability of proteins to form stable emulsions between immiscible oil and water phases. This is a key property enabling the stabilization of oil–water emulsions in a wide range of food products [12]. The obtained values of emulsification activity (EA) and emulsion stability (ES) are presented in Figure 2.

Among the analyzed protein preparations, the highest emulsifying activity (EA ≥ 58%) was demonstrated by pea protein isolates PP1, PP2, PP3, and PP4, as well as rice proteins RP1 and RP2, which are isolates with a protein content above 80%. Equally high results were obtained for chickpea (CP1) and oat (OP1) protein concentrates, indicating their good ability to stabilize biphasic systems despite their lower total protein content. High EA values indicate the high ability of protein molecules to quickly reduce the interfacial tension and their efficient adsorption at the phase boundary, which is crucial in the initial stage of emulsion formation [48].

In the case of preparations HP1, HP2 (hemp proteins), and SP1 (sunflower), a significantly lower EA of 5–20% was observed, which may indicate the limited ability of these proteins to adsorb at the phase interface and form stable surface layers. This may be due to the presence of large amounts of insoluble fractions or protein–lipid complexes, which hinder the unfolding of the protein structure and reduce the surface activity of the proteins [49,50].

A similar relationship was observed for ES. High ES (≥58%) was observed in the same samples with the highest EA. EA significantly correlated with ES, indicating that the ability to initiate emulsion formation is closely linked to its subsequent stability, and both parameters reflect common surface mechanisms. This suggests favorable structural properties of these proteins, such as amphiphilicity and the ability to form flexible and shear-resistant interfacial films that effectively prevent oil droplet coalescence.

Hemp and sunflower proteins demonstrated significant phase separation shortly after emulsion preparation, which translated into low ES (<10%).

The observed differences can be partially explained by the degree of purification and the molecular structure of individual proteins. Pea isolates are characterized by higher purity, better solubility, and a higher proportion of globulin fractions (including legumin and vicilin), which demonstrate good ability to form stable emulsions [51]. The high emulsifying activity demonstrated by emulsions stabilized with pea protein isolates has been confirmed in numerous scientific studies [52,53,54]. In turn, hemp and sunflower preparations often contain insoluble fiber fractions, polyphenols, or lipid–protein complexes, which may limit their technological functionality. Such molecules may compete with proteins for adsorption sites or create unstable surface structures, which significantly impairs their emulsion functionality. Even after various isolation methods, the emulsification of these proteins is poor, and the emulsion stability is limited [55,56].

The high emulsifying capacity of chickpea and oat protein concentrates may be related to the presence of natural emulsifiers, such as polysaccharides (β-glucans) or saponins, which support the stability of the system. Fine β-glucan particles can completely cover the surface of the oil droplets, creating a physical barrier against aggregation [57,58].

### 3.4. Storage Stability of Emulsions (SP)

Emulsion stability is an important factor influencing the shelf life of emulsified foods. Emulsions can be destabilized by several mechanisms, such as creaming (upward movement of droplets due to density differences), flocculation (reversible formation of droplet aggregates without their merging), and coalescence (irreversible merging of droplets leading to an increase in their diameter). These processes occur in parallel or sequentially and significantly accelerate phase separation, especially in emulsions stabilized with weaker emulsifiers [59,60,61]. The results are presented in the graph (Figure 3) as a graphical summary of stability changes over time, as well as in photographs (Figure 4), which illustrate the visual differences in emulsion appearance observed during subsequent days of storage.

Of all the protein preparations analyzed, sunflower protein (SP1) demonstrated the highest SP value throughout the study period. Just 3 h after emulsion preparation, its stability reached 80.00 ± 0.00%, and after 6 days, it remained at a very high value of 74.00 ± 0.14%. This may indicate the favorable surface and structural properties of this preparation, potentially related to the presence of natural phospholipids and cell membranes, which support the formation of the interfacial layer and prevent coalescence of fat droplets [62]. As demonstrated in the study by Pérez-Gálvez et al. [63], sunflower protein hydrolysates can act as effective emulsifiers, ensuring high physical stability of emulsions.

Emulsions based on rice (RP1, RP2), pea (PP1-PP4), chickpea (CP1), and oat (OP1) proteins demonstrated similar levels of storage stability, reaching 65–70% on the day of preparation. Minor differences were observed on subsequent days, but in most cases, these systems demonstrated relatively good emulsion phase retention even after 6 days of refrigerated storage (4 °C). Unmodified plant proteins can stabilize emulsions for short periods, but their long-term storage stability is limited, as most emulsions tend to separate and flocculate during storage [64]. Ho et al. [65] showed that emulsions based on pea protein isolates remain physically stable (no separation or changes in droplet size) for about 14 days at 4 °C (pH 7), but their stability is lower than that of emulsions with milk proteins.

Storage stability (SP) studies confirmed the poor functional properties of hemp proteins. In the case of sample HP2, the SP value was only 55.00 ± 0.28% after 6 days of refrigerated storage. These results indicate the limited ability of hemp proteins to maintain emulsions in a physically stable state for extended periods. This is mainly due to their poor solubility, tendency to aggregate, and limited ability to form a stable interfacial layer. Similar results were obtained in studies by Dapčević-Hadnađev et al. [50] and Feng et al. [66]. In this case, emulsions stabilized solely with hemp protein were characterized by rapid coalescence and a significant increase in droplet size. Under accelerated storage conditions (55 °C), an increase in the average droplet size of up to 50 times was recorded within 9 days, indicating the limited ability of hemp protein to maintain the stability of the emulsion system.

### 3.5. Color Determination of Plant Emulsions Using the CIE L*a*b* Model

Color is one of the key characteristics that determine consumer perception of a product—particularly in instant products and milk substitutes, where a light, creamy color is associated with purity, freshness, and high quality [67]. The values of the *L**, *a**, and *b** color parameters obtained for the tested plant emulsions are presented in Table 3.

The *L** (lightness) parameter values ranged from 48.07 ± 0.15 (HP2) to 73.22 ± 0.57 (RP2). The lightest emulsions were obtained with rice protein RP2, pea proteins (PP1, PP2), and chickpea protein (CP1), making them attractive for use in products with a desired “milky” color. The obtained results are consistent with reports by other authors, who indicate that the light colors of emulsions based on plant proteins—particularly rice and pea proteins—result from the low content of natural pigments and a favorable amino acid profile, favoring the formation of stable, light colloidal systems [68,69]. Hemp proteins (HP1 and HP2) produced significantly darker emulsions with a reddish-brown color, which may limit their use in products where high visual aesthetics are expected.

The *a** parameter (red–green color index) was positive for most samples, with the highest values obtained for hemp emulsions (HP2: 2.80 ± 0.07; HP1: 2.02 ± 0.03), which may indicate the presence of color compounds derived from cell membrane debris, chlorophyll, or polyphenols [70]. In turn, samples with the lowest *a** values (below zero)—such as PP2, PP3, PP4, and CP1—showed a slight tendency towards cooler (greenish) colors, which is beneficial for imitating plant-based dairy products [68].

The *b** parameter (responsible for the blue–yellow color) was positive in all samples, indicating a general creamy-yellow color of the emulsion, typical of plant products. The highest *b** value was obtained for the rice protein sample RP1, and the lowest for the pea protein sample PP2.

The *L** parameter was observed to be positively correlated with OAC, while the *a** parameter correlated with both WAC and OAC. These correlations may be due to the fact that lighter protein preparations are usually better purified and have lower pigment and insoluble fraction content, which favors their greater surface accessibility and reactivity towards oil and water phases, and thus higher WAC and OAC values.

Moreover, significant positive correlations were found between the color parameter *L** and the functional properties FC, FS, EA, and ES (r = 0.6979, 0.6138, 0.8816, and 0.8943, respectively). This indicates that the lighter the sample (and therefore most likely better purified and with a lower content of pigments and interfering fractions), the greater the ability to form foams and emulsions and stabilize them. This is consistent with surface mechanisms, whereby purer formulations create more homogeneous, stable interfacial films.

The color differences (Δ*E*) between the tested emulsions and the standard (white calibration plate) ranged from 16.93 to 40.18, indicating clearly noticeable, and in some cases very large, color deviations (Table 3).

The lowest Δ*E* values were obtained for samples RP2 (Δ*E* = 16.93), PP1 (Δ*E* = 18.16), PP2 (Δ*E* = 17.29), and CP1 (Δ*E* = 18.38). This means that emulsions based on these proteins differed the least in terms of color from the white standard, which may indicate higher brightness and a more uniform shade resulting from lower intensity of red and yellow colors. This color characteristic is desirable in the context of food applications, especially products such as coffee creamers. The highest color differences relative to the white standard were observed for emulsions containing hemp protein HP1 (Δ*E* = 37.52) and HP2 (Δ*E* = 40.18). Hemp proteins may require additional technological treatments (e.g., microfiltration, bleaching, decolorization) if they are to be used in products with high aesthetic requirements [71]. Such differences can affect the visual characteristics of formulated emulsions and should be considered when selecting protein ingredients for food applications, especially products with a desired light color.

Based on the obtained results (3.1–3.5), including emulsification activity and stability (EA and ES), ability to form and maintain foam (FC and FS), sorption properties (WAC and OAC), and color parameters (CIE *Lab** model), a preliminary selection of the most promising protein preparations was made for further research. The main qualifying criterion was high emulsification potential—particularly the ability to form stable emulsions (high EA and ES values), complemented by the ability to form and stabilize foam, which may indicate good protein surface activity. At this stage, hemp proteins (HP1 and HP2), oat protein (OP1), and sunflower protein (SP1) were excluded, as they—despite certain technological advantages—exhibited relatively low functionality in emulsion systems and unfavorable sensory characteristics (e.g., color and foamability). Pea isolates, rice proteins and chickpea proteins were selected for further analysis, as they combined favorable technological properties with high visual quality of the emulsion.

### 3.6. Determination of the Viscosity of Protein Solutions

The dynamic viscosity values of 5% solutions of the analyzed proteins are summarized in Figure 5.

As previously demonstrated by González-Tello et al. [72], the viscosity of whey protein solutions at low concentrations (<20%) is not a function of shear rate, so such solutions exhibit Newtonian fluid properties, which may be the result of the decomposition of aggregated proteins during shear. The viscosity of protein solutions is determined by their concentration, molecular weight, size and shape of protein molecules, degree of hydration, and intermolecular interactions [72]. The variation in viscosity of the tested protein solutions was rather limited. Samples RP1, RP2, CP1, and PP2 exhibited similar viscosity. Samples PP3 and PP4 were slightly higher. However, the pea protein sample PP1 exhibited more than twice the viscosity compared to the others. This may be due to the presence of additional non-protein components, such as high-molecular-weight, branched polysaccharides, which significantly increase solution viscosity. Importantly, sample PP1 had the highest WAC and the lowest zeta potential, suggesting a higher degree of hydration and stronger intermolecular interactions, which may contribute to the increased viscosity. It was also observed that the viscosity of protein solutions significantly (*p* < 0.05) correlated positively with WAC (r = 0.8737) and WOAI (r = 0.8905). This indicates that proteins with a greater ability to bind water form more viscous systems because more intense hydration and stronger interactions between hydrated molecules increase flow resistance.

According to González-Tello et al. [72], the viscosity of whey protein solutions at 25 °C ranges from 1.84 to 2.63 mPa·s. Krstonošić et al. [73] showed that soy and pea protein solutions are characterized by greater relative lightness compared to soy protein solutions.

### 3.7. Determination of Surface Tension

Protein molecules have the ability to lower surface and interfacial tension due to their surface-active nature. The surface tension values of the tested protein solutions are summarized in Figure 6.

The surface tension of pure water is 72 mN/m. The presence of protein molecules in water significantly reduced the surface tension in each case. The lowest surface tension values were observed for solutions of RP2 and PP1 proteins, below 45 mN/m, indicating their highest surface activity. In contrast, CP1 protein reduced the surface tension of water to the lowest degree, with the solution having a value of approximately 55 mN/m. As demonstrated by González-Tello et al. [72], the surface tension of whey protein solutions is 42.5 mN/m, and a concentration of 5% is the critical value above which this parameter does not change significantly. Furthermore, the cited authors found that the surface tension of whey protein solutions decreases linearly with temperature. Similar surface tension values ranging from 44.2 mN/m for potato protein to 46.0 mN/m for whey proteins were determined by Krstonošić et al. [73].

### 3.8. Zeta Potential Determination

Surface charge (zeta potential) reflects variations in protein composition and structure and influences solubility and emulsification properties. Surface charge can be influenced by several diverse factors, including pH, chemical composition, temperature, and concentration. The presence of ionic groups (e.g., -COO^−^ and -NH^3+^), hydrophilic polar groups (e.g., -NH_2_ and -OH), and nonpolar hydrophobic residues, e.g., alkyl and aromatic groups, influences the surface charge [74]. A higher absolute zeta potential (more negative or more positive) suggests greater electrostatic repulsion, reducing the likelihood of aggregation and, therefore, greater system stability [75]. The zeta potential values of the studied proteins are shown in Figure 7.

The lowest values, below −35 mV, were obtained for samples PP1, PP2, and PP4. The highest, −24.62 mV, was obtained for the RP1 protein. Statistical analysis did not reveal any significant (*p* < 0.05) correlations between zeta potential and emulsifying properties (EA and ES) or storage stability (SP), suggesting that the stability of these systems depended to a greater extent on other functional properties of proteins, such as the ability to form an interfacial film or the presence of non-protein components. As shown by Tang et al. [74], the zeta potential of different protein isolates showed a similar pattern: initially positive at pH 3.0 and gradually decreasing to zero at the pH in the isoelectric point region (pH 4–5), and then showing a negative trend with further increase in pH in the range 5–11. This is due to the ionization of amino acids, which affects the distribution of charged groups depending on pH, causing carboxyl groups to be protonated under acidic conditions and amino groups to be deprotonated under alkaline conditions. The zeta potential values determined by the cited authors for different protein isolates ranged from approximately 30 mV at pH 3.0 to almost −40 mV at pH 9.0, with the lowest values found for soy protein. Similarly, da Silva et al. [54] observed a significant dependence of the zeta potential of plant proteins on pH. At low pH values, pea protein showed the highest zeta potential (approximately 30 mV), whereas at higher pH (>6.0), rice protein had the lowest value of this parameter (approximately −30 mV). Olatunde et al. [75] determined the zeta potential value for yellow pea protein and modified preparations obtained from it in the range of −23.53 to −18.33 mV.

## 4. Conclusions

The analyzed plant protein preparations differed notably in their physicochemical and functional properties. Pea (PP1–PP4), rice (RP1, RP2), and chickpea (CP1) proteins demonstrated the most favorable properties, showing high emulsifying activity, good emulsion stability, and advantageous color and surface properties. Although chickpea protein is formally classified as a protein concentrate, it exhibited functional properties comparable to the isolates. Significant correlations were observed between *L** values and OAC, FC, FS, EA, and ES, as well as between viscosity and WAC and WOAI. The overall results indicate that these proteins, particularly pea, rice, and chickpea, represent promising candidates for incorporation into plant-based emulsion systems.

Future work will focus on using the selected proteins to design optimized plant-based emulsions and evaluate their technological suitability in more advanced applications.

## Figures and Tables

**Figure 1 foods-14-04291-f001:**
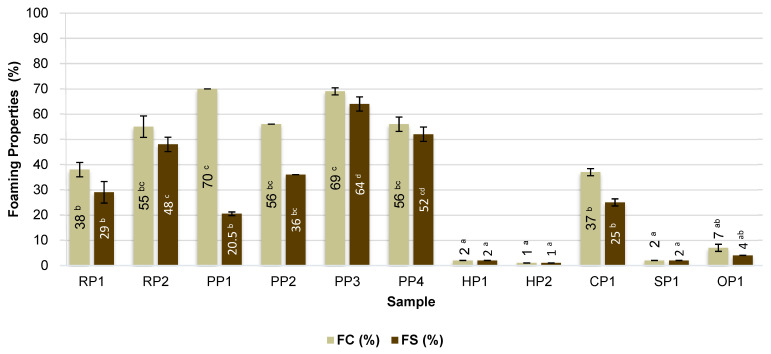
Foaming properties of commercial plant-based protein preparations. RP—rice proteins, PP—pea proteins, HP—hemp proteins, CP—chickpea protein, SP—sunflower protein, OP—oat protein, FC—foaming capacity, FS—foaming stability. Different superscripts within the same parameter (FC or FS) followed by different letters (^a–d^) indicate a significant difference (*p* < 0.05).

**Figure 2 foods-14-04291-f002:**
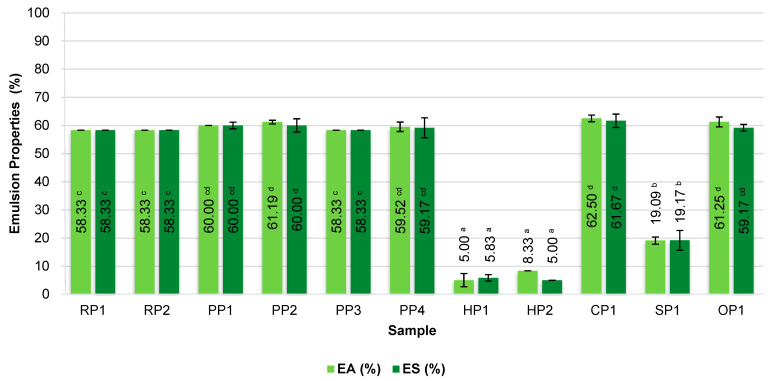
Emulsification of commercial plant-based protein preparations. RP—rice proteins, PP—pea proteins, HP—hemp proteins, CP—chickpea protein, SP—sunflower protein, OP—oat protein, EA—emulsification activity, ES—emulsification stability. Different superscripts within the same parameter (EA or ES) followed by different letters (^a–d^) indicate a significant difference (*p* < 0.05).

**Figure 3 foods-14-04291-f003:**
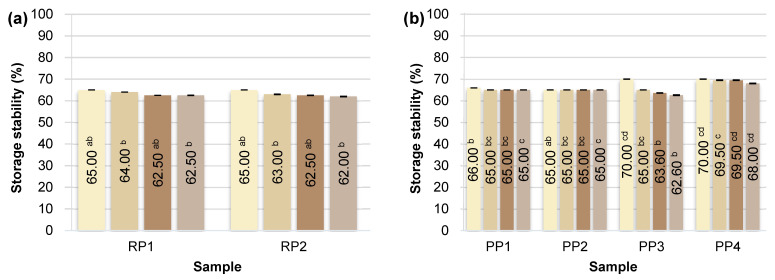
Storage stability of plant emulsions after 3 h and 2, 4, 6 days of storage at 4 °C: (**a**) RP—rice proteins; (**b**) PP—pea proteins; (**c**) HP—hemp proteins; (**d**) CP, SP, OP—chickpea, sunflower and oat proteins. Superscript letters indicate statistically significant differences among protein samples within the same storage time (*p* ≤ 0.05). Statistical analysis was performed separately for each time point.

**Figure 4 foods-14-04291-f004:**
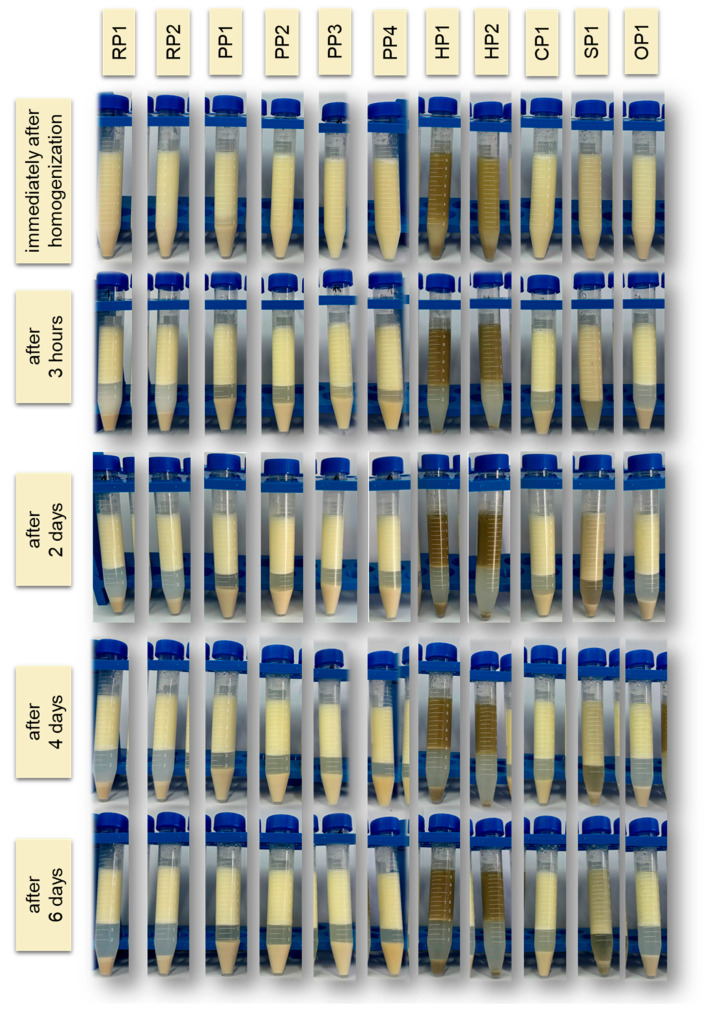
Visual changes in the emulsion observed immediately after homogenization and after 3 h, 2, 4 and 6 days of storage at 4 °C. RP—rice proteins, PP—pea proteins, HP—hemp proteins, CP—chickpea protein, SP—sunflower protein, OP—oat protein.

**Figure 5 foods-14-04291-f005:**
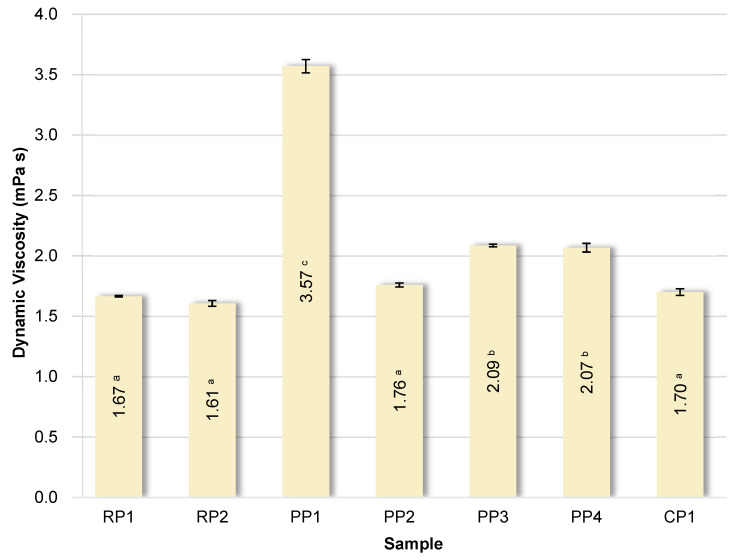
Dynamic viscosity of solutions of analyzed proteins. RP—rice proteins, PP—pea proteins, CP—chickpea protein. Data marked with different letters are significantly different at *p* ≤ 0.05.

**Figure 6 foods-14-04291-f006:**
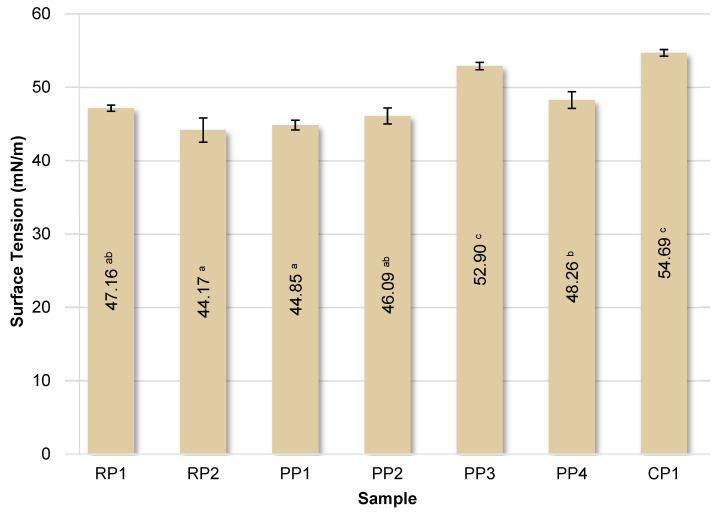
Surface tension of the analyzed protein solutions. RP—rice proteins, PP—pea proteins, CP—chickpea protein. Data marked with different letters are significantly different at *p* ≤ 0.05.

**Figure 7 foods-14-04291-f007:**
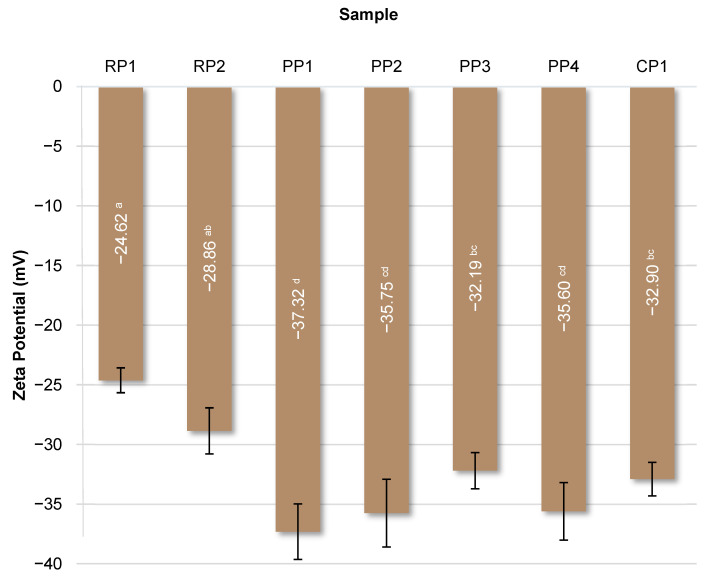
Zeta potential of analyzed proteins. RP—rice proteins, PP—pea proteins, CP—chickpea protein. Data marked with different letters are significantly different at *p* ≤ 0.05.

**Table 1 foods-14-04291-t001:** List of plant proteins used with codes.

Protein Type	Code	Company	Type	Protein (g/100 g) *	Fat (g/100 g) *	Carbohydrate (g/100 g) *	Fiber (g/100 g) *
Rice protein (RP) 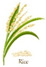	RP1	Kerry Group (Tralee, Ireland)	isolate	about 80	3	12.5	n.a **
RP2	Falken Trade (Olsztyn, Poland)	isolate	≥80	6	n.a.	n.a.
Pea protein (PP) 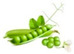	PP1	Kerry Group (Tralee, Ireland)	isolate	about 80	4.6	3	n.a.
PP2	Falken Trade (Olsztyn, Poland)	isolate	≥80	≤10	n.a.	≤0.5
PP3	Roquette Frères (Lestrem, France)	isolate	≥83	n.a.	n.a.	10
PP4	Ingredion (Hamburg, Germany)	isolate	≥84	7.5	0.3	2.6
Hemp protein (HP) 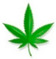	HP1	ProAgro GmbH (Wien, Austria)	concentrate	56	4.5	8.2	16
HP2	BioPlanet S.A. (Wilkowa Wieś, Poland)	concentrate	49	11	4.2	24
Chickpea protein (CP) 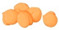	CP1	Innovopro (Ra’anana, Israel)	concentrate	70	10.6	5.7	5.1
Sunflower protein (SP) 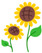	SP1	BioGol (Trzebiatów, Poland)	concentrate	50	19	4.6	11
Oat protein (OP) 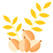	OP1	Helhetshälsa (Borghamn, Sweden)	concentrate	53	13	21.7	4.6

* Data based on product specification sheets provided by manufacturers. ** n.a.—not available.

**Table 2 foods-14-04291-t002:** The ability of selected plant proteins to absorb water and oil and WOAI index.

Sample	Water Absorption Capacity, WAC [g H_2_O/g Protein]	Oil Absorption Capacity, OAC [mL Oil/g Protein]	Water–Oil Absorption Index, WOAI Index g H_2_O/mL Oil
RP1	2.97 ± 0.20 ^b^	2.10 ± 0.01 ^D^	1.41
RP2	3.18 ± 0.24 ^b^	2.10 ± 0.01 ^D^	1.51
PP1	6.81 ± 1.03 ^c^	2.10 ± 0.75 ^D^	3.24
PP2	3.97 ± 0.01 ^b^	2.00 ± 0.28 ^C^	1.99
PP3	4.66 ± 0.16 ^c^	2.40 ± 0.01 ^E^	1.94
PP4	5.50 ± 0.76 ^c^	2.15 ± 0.07 ^D^	2.56
HP1	2.44 ± 0.25 ^a^	1.90 ± 0.01 ^C^	1.28
HP2	2.44 ± 0.02 ^a^	1.70 ± 0.01 ^A^	1.44
CP1	3.39 ± 0.13 ^b^	2.45 ± 0.07 ^E^	1.38
SP1	3.87 ± 0.58 ^b^	1.80 ± 0.01 ^B^	2.15
OP1	3.67 ± 0.54 ^b^	1.95 ± 0.07 ^C^	1.88

RP—rice proteins, PP—pea proteins, HP—hemp proteins, CP—chickpea protein, SP—sunflower protein, OP—oat protein. Values are mean ± standard deviation. Data marked with different letters are significantly different at *p* ≤ 0.05. Lowercase letters denote statistically significant differences among samples within the same column, while uppercase letters also indicate statistically significant differences among samples within the same column.

**Table 3 foods-14-04291-t003:** Color parameters (*L**, *a**, *b**), color difference (Δ*E*) and visual appearance of plant emulsions prepared using different proteins.

Code	*L**	*a**	*b**	Δ*E*	Appearance
RP1	69.80 ± 0.34 ^d^	2.70 ± 0.56 ^cd^	17.88 ± 0.52 ^d^	22.95	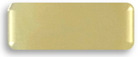
RP2	73.22 ± 0.57 ^e^	0.63 ± 0.41 ^b^	13.02 ± 0.77 ^b^	16.93	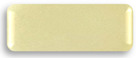
PP1	70.32 ± 0.20 ^d^	0.08 ± 0.24 ^b^	11.34 ± 0.76 ^ab^	18.16	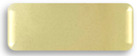
PP2	70.49 ± 0.25 ^d^	−0.35 ± 0.27 ^ab^	9.95 ± 0.64 ^a^	17.29	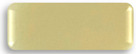
PP3	69.05 ± 0.59 ^d^	−0.74 ± 0.12 ^a^	11.98 ± 0.62 ^ab^	19.58	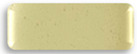
PP4	69.03 ± 0.24 ^d^	−0.57 ± 0.01 ^a^	12.44 ± 0.30 ^ab^	19.85	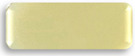
HP1	51.09 ± 0.17 ^b^	2.02 ± 0.03 ^c^	16.16 ± 0.17 ^c^	37.52	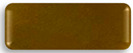
HP2	48.07 ± 0.15 ^a^	2.80 ± 0.07 ^d^	15.68 ± 0.19 ^c^	40.18	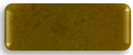
CP1	70.45 ± 0.10 ^d^	−0.74 ± 0.01 ^a^	11.92 ± 0.27 ^ab^	18.38	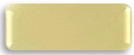
SP1	66.83 ± 0.44 ^c^	0.21 ± 0.01 ^b^	11.35 ± 0.10 ^ab^	21.17	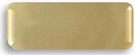
OP1	67.24 ± 0.64 ^c^	0.74 ± 0.23 ^b^	12.56 ± 0.28 ^ab^	21.43	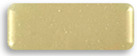

RP—rice proteins, PP—pea proteins, HP—hemp proteins, CP—chickpea protein, SP—sunflower protein, OP—oat protein. Different superscript letters within each column indicate statistically significant differences among protein samples (*p* ≤ 0.05). Statistical comparisons were performed independently for each color parameter (*L**, *a**, *b**).

## Data Availability

The original contributions presented in the study are included in the article, further inquiries can be directed to the corresponding author.

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
