# Peer review of "Plant Proteins as Alternative Natural Emulsifiers in Food Emulsions"

_foods, 2025, doi:10.3390/foods14244291_

Round 1

Reviewer 1 Report

Comments and Suggestions for Authors

In this study, the authors examine the primary emulsifying properties and selected technological characteristics of protein isolates obtained from various plant sources. Considering the increasing interest in and demand for vegan protein ingredients, the topic is timely and has the potential to contribute meaningfully to the existing literature.

For a high-quality Q1 journal such as Foods, it would be beneficial for the manuscript to include a more comprehensive and in-depth analysis of the characterized parameters, accompanied by a more detailed and critically oriented discussion. Strengthening these sections would considerably enhance the scientific robustness and overall presentation of the work.

Overall, the manuscript is written in clear and understandable language, and the study offers valuable findings that may guide future research in the field. However, there are several important issues that should be carefully addressed to improve the quality and clarity of the manuscript. Therefore, I recommend that the manuscript be reconsidered for publication after a major revision.

My specific comments and suggestions are as follows:

Title: If the analyses are revised within the suggested framework and the findings are strengthened through a more comprehensive emulsion characterization, the manuscript’s title can be considered appropriate

Keywords: For the keywords, instead of repeating terms from the title, I recommend selecting words that highlight other characterized properties or key findings of the study, which will also improve discoverability and indexing in a broader context

Introduction: Although the study appears sufficient in terms of the characterized properties, the overall evaluation remains rather superficial. In particular, the relevance of the spray-drying process highlighted in the Introduction is unclear, as its connection to the experimental work is not adequately established. The manuscript characterizes the primary emulsion properties of 11 plant-based protein samples, along with certain technological properties supporting emulsification; however, the link between these analyses and the stated focus on spray drying needs to be clarified.

# To deepen the study, the emulsion properties could be evaluated at different pH levels, and the potential application of these protein sources in various product categories could also be examined by assessing their emulsifying behavior under different conditions

# When justifying the significance of the study, the Introduction could be strengthened by more clearly emphasizing the growing need for plant-based emulsifier sources and how these can be improved through various modification techniques, particularly enzymatic hydrolysis. Incorporating recent studies supporting these points would help reinforce the rationale. For example, the study I suggested can be used to expand and update this section. (Line 52-57)

 https://www.sciencedirect.com/science/article/pii/S0141813025093808?casa_token=Kk4LuLDaedEAAAAA:mWnPwrHdjNhH2hNfhSSDAfhELL8IJbBcqpP24008lNG7XWLubPJazWmOfxWl1zIj7MZ08yncfg

Material and Methods:

# The Materials section is overly long and contains several repetitive statements. For example, it is sufficient to state that the samples were protein isolates and concentrates without any applied modification processes, rather than repeating similar descriptions multiple times.

# Please ensure that the same units are used consistently for all characterized properties in both the Methods section and the corresponding tables and figures in the Results.

# For the color characteristics, calculating and reporting the ΔE (overall color difference) values would further strengthen the analysis by clearly demonstrating the differences between the samples in terms of their general color properties.

# When reporting instrument or materail details, it would be more appropriate to provide the model information in parentheses (Line 105 MPW-380, Line 119   Wielkopolski, Line 137 IKA, Line 155 Wielkopolski,Line 157 MPW-380, Line 182 Chroma Meter CR-400 , Line 193 etc.)

# Please check the units throughout the manuscript and ensure that a consistent formatting style is used.please correct the statement( Line 187 5 g/100 mL (m/v) as  (w/v), Line 19 (m/v) )

# Please check the units reported in Table 2 and ensure correct and consistent usage. For example, the unit for Water absorption capacity (WAC) should be expressed as g H₂O/g protein or g H₂O/g sample.

# Although the tables and figures sufficiently present the data, they require further refinement and should be formatted using a consistent and standardized writing styl. In this context, please review all tables and figures carefully and apply the necessary revisions.( especially x-y axis titles)

# Please prepare the table and figure titles and notes more carefully and in a consistent style. For example, when explaining abbreviations in the notes, avoid using terms like ‘explanations’ and instead present them directly, such as ‘ES: emulsion stability,’ and apply the same formatting to all other abbreviations

# Similarly, please check and correct the units for oil absorptioncapacity (OAC) and the other reported parameters to ensure accuracy and consistency.

# Please ensure a consistent writing style for all labels, titles, and captions in the tables and figures. Use either title case (each word capitalized) or sentence case (only the first word capitalized) throughout (Table 2- Water absorption capacity, Oil absorption capacity, Water-Oil Absorption Index, WOAI index ).

Good luck with corrections…

Author Response

Dear Reviewer,

Please find attached word file with our answers to your's valuable comments and suggestions.

Reviewer 2 Report

Comments and Suggestions for Authors

In this study, the authors have done commendable work, and the results shown are promising. However, there are numerous grammatical errors, unit errors, and improper explanations of methodology, as well as equations that require corrections. Additionally, the concept of this study is still uncertain, as it compares protein isolate from one source to protein concentrate from another.

Detailed corrections are provided below.

  • Abstract needs to be rewritten with some important data.
  • Table 1: On what basis are the plant proteins selected? Also, why do some plant proteins have isolates selected, whereas others have concentrates? Why not both for each type, or only isolates or concentrates of each one?
  • The study is a comparison of protein isolates from some plant sources to protein concentrates from another plant source. Comparison should be within the isolate and within the concentrate for better understanding.
  • Use standard units throughout the manuscript. For example, in line 103, “minutes” is written, whereas in line 105, “min” is written. It should be consistent throughout the manuscript.
  • Line 110: H2O is written incorrectly.
  • In Eq. 1, clearly mention W, weight of what, weight of water added, or weight of protein in the sample, or weight of the sample initially or afterwards. So, it should be clearly mentioned.
  • Line 135: Maintain consistency; somewhere foaming capacity is written, somewhere foaming efficiency is written. It should be consistent throughout the manuscript.
  • Check line 153: “To 15 mL of distilled water ….. Wielkopolski rapeseed oil”. The meaning is unclear.
  • Line 159: The terms used in equation 5 should match the text, even to subscript or superscript.
  • Table 2: Check the unit of oil absorption capacity; it should be mL of oil absorbed per gram of sample, which is written incorrectly. Based on that, WOAI will also change now. Kindly recalculate and check.
  • Line 232: The unit is written as gram H2O per gram protein, whereas in line 227, it is not clearly mentioned. The unit should be consistent throughout and clearly stated.
  • Similarly, for OAC, it is mL/g; it should be properly written as mL of oil per g of protein.
  • No proper reason is provided for analyzing the properties of protein isolate samples and omitting concentrates for viscosity, surface tension, and zeta potential.
  • Reasoning for the data is missing at various places, which needs to be checked and added.
  • In the conclusion section, there are no concluding remarks. It is a basic phenomenon that protein isolates exhibit better emulsifying and foaming properties, as well as greater stability, compared to concentrates. For that, no such analysis or study is required. Please revise the conclusion to include some significant findings from the results.

Addressing these concerns will definitely improve the manuscript and make it more suitable for publishing in such journals.

Author Response

(The authors gave the same response as above.)

Reviewer 3 Report

Comments and Suggestions for Authors

The manuscript investigates functional properties of several commercial plant proteins and evaluates their suitability as natural emulsifiers for food emulsions, particularly instant products that undergo spray-drying. The topic is timely, fits well within current clean-label and plant-based product trends, and the manuscript is generally well structured.

However, there are areas where clarity, methodological justification, depth of discussion, and statistical rigor could be strengthened.

ABSTRACT

  • Summarizes purpose, methods, and principal findings well.

Weak Points

  • Overstates novelty (“confirms the potential…”), without showing thresholds or benchmarks.
  • Mentions spray-drying suitability although no spray-drying experiments are included.
  • Lacks quantitative values-adding representative data would strengthen it.

1.INTRODUCTION

  • Provides strong context for the need for natural, plant-based emulsifiers and is well supported by literature; broad overview of protein digestibility and amino acid profiles. Also, clearly states the objective and links research to industrial applications (instant products, spray drying).

As weak points I can mention:

  • Too long; includes tangential nutrition content not directly linked to emulsification experiments.
  • Claim of novelty (“no studies linking such broad characteristics…”) needs stronger justification.
  • Spray-drying emphasis is strong, but no spray-drying data is presented later.
  • Introduction would benefit from a clearer description of the knowledge gap.

2.MATERIALS and METHODS

As strong points: Table 1 is informative, includes protein content, classification, source, and manufacturer. Clarifies that all proteins were used in unmodified form (good methodological control).

As regarding the Methods, all experimental protocols are clearly described and properly referenced. Replication (triplicate) is specified. Covers a wide set of functional metrics relevant to emulsification. Equations and operational steps are well structured.

Weak Points: No rationale provided for protein selection (e.g., why no soy protein?). No proximate composition (fat, fiber, ash) provided; this limits interpretation of differences.

  • pH of protein solutions is not reported; pH strongly affects solubility and emulsification.
  • Ionic strength not controlled or reported.
  • No assessment of protein solubility, despite discussing solubility properties in the text.
  • Storage stability study (6 days) is not justified for an emulsion intended for spray drying (short-term use).
  • Statistical methods section is vague (using Excel + Statistica + SPSS—too many tools listed).
  • No mention of controlling particle size or filtration before measurement.

3.RESULTS & DISCUSSION

3.1 Water Absorption Capacity (WAC) & Oil Absorption Capacity (OAC)

Strong Points: Data clearly presented in Table 2. Good interpretation linking WAC and OAC to technological applications. WOAI index explanation is useful.

Weak Points: Explanations about fiber content in hemp are speculative—no composition data provided. Discussion lacks statistical depth (no p-values shown). Not linked to actual emulsion outcomes (correlation analysis missing).

  • Foaming Properties

Strong Points: Good connection to protein structure (albumins vs globulins). Identification of pea and rice proteins as good foamers supported by literature. Data presented clearly with statistical superscripts.

Weak Points: Foam stability is only measured at 10 minutes; no longer-term evaluation. No solubility data to support claims about why hemp proteins perform poorly. Limited mechanistic explanation (e.g., interfacial film elasticity not discussed).

  • Emulsifying Activity (EA) & Emulsion Stability (ES)

Strong Points: Well-structured results showing strong performance of pea, rice, chickpea proteins. Interpretation aligned with known functional behavior of globulins and albumins.

Weak Points: Emulsions are prepared at only one protein concentration; no dose-response evaluation. ES experiment uses very mild heat treatment (30°C), which does not simulate real product processing. No droplet size measurements to substantiate discussions about stability. Discussion sometimes repeats results instead of explaining mechanisms.

  • Emulsion Storage Stability (SP)

Strong Points: Provides both numerical summary and visual evidence (Figures 3 & 4). Systematically compares stability over 3h, 2d, 4d, 6d. Identifies sunflower protein as highly stable, which is interesting.

Weak Points: This section yields a contradiction: sunflower protein performs best here but was excluded from “promising candidates” without explanation later. No mention of physical instability mechanisms (creaming vs coalescence vs flocculation). No particle size measurements to support claims. Storage study not highly relevant for emulsions intended to be spray-dried shortly after creation.

3.5 Color Measurements (CIE Lab)*

Strong Points: Data presented well in Table 3. Correctly highlights proteins with the most desirable color (RP2, PP1/PP2, CP1). Discussion ties color to consumer perception.

Weak Points: No sensory validation or comparison with real dairy creamer standards. Discussion of color chemistry (pigments, Maillard reactions, polyphenols) is superficial. Lack of correlation with WAC/OAC or emulsification performance.

  • Viscosity

Strong Points: Relevant to processing behavior; viscosity directly affects spray-drying feasibility. Identification of unusually high viscosity for PP1 is interesting.

Weak Points: No explanation for why PP1 viscosity is twice as high as others—composition not analyzed. No measurement across shear rates (assumption of Newtonian behavior may not hold). No link between viscosity and emulsification outcomes.

  • Surface Tension

Strong Points: Important metric for protein functionality in emulsions. Correctly interpreted: lower surface tension indicates higher interfacial activity.

Weak Points: No time-dependent adsorption behavior measured (dynamic surface tension). No correlation between surface tension and emulsification activity, though this is expected.

3.8 Zeta Potential

Strong Points: Relevant for understanding electrostatic stabilization. Correct interpretation: more negative values improve stability.

Weak Points: Only one pH tested; zeta potential profiles across pH range would be more informative.

No droplet zeta potential—only protein solution measured. No link to ES or SP data.

4.CONCLUSIONS

Strong Points: Clear summary of main functional findings. Identifies most promising proteins for further work (pea, rice, chickpea). Relates findings to clean-label and plant-based trends.

Weak Points: Overstates application readiness (“real prospects for implementation”). Does not acknowledge contradictions (e.g., sunflower protein stability). No limitations or future work section included. Spray-drying claim is unsupported by actual experimental data.

Author Response

(The authors gave the same response as above.)

Round 2

Reviewer 1 Report

Comments and Suggestions for Authors

The authors have addressed the revisions carefully and thoroughly, implementing the requested changes and substantially improving the quality of the manuscript. In particular, the revised keywords are now more appropriate and will enhance the discoverability of the study within a broader scientific literature. In the introduction, the authors have successfully integrated previous and recent findings related to the investigated characteristics and have clearly articulated the purpose of the study.

With these improvements, the manuscript has become much stronger and can be considered of publishable quality. However, before final acceptance, I would like to draw attention to a few minor points that appear to have been overlooked. I recommend that these issues be corrected, after which the manuscript can be accepted for publication.

#Line 227-310-. Please ensure that the color parameters L*a*, and b*—which represent the measured color values—are consistently written in italics throughout the manuscript. This correction should be applied particularly in the title and in all relevant sections of the text to maintain proper scientific notation and formatting.

#Line 495- 567. Please check the manuscript thoroughly and ensure that the color parameters L*a*, b* and ΔE are consistently formatted in italics throughout the text. This correction should also be applied to the title and any other relevant sections to maintain accurate and standardized scientific notation. And Table 3 and conclusion section.

#In Figure 2, the axis label currently written as “Emulsification [%]” may be corrected to “Emulsion Properties (%)” for improved clarity and accuracy

#As mentioned in my previous revision comments, please ensure that each table and figure includes the abbreviations indicating the protein sources used. For example, Pea Protein (PP), Hemp Protein (HP), and the corresponding abbreviations for the remaining protein types should be clearly defined and presented in the captions.

# Line 185, 248, 256 and etc. Please ensure consistent notation for room temperature throughout the manuscript. A standardized expression such as “approximately 25 °C” or simply  “ ~22 °C” should be used uniformly in all relevant section. I think correct as   ~22 °C in related part.

Line 277 and Table 2. In Table 2, the asterisk (*) appears to refer only to the Water Absorption Capacity value of RP1, which may create ambiguity. I recommend removing the asterisk and instead adding a general statement below the table indicating that all values are presented as mean ± standard deviation. This would improve clarity and consistency in data presentation.

# Please add a note below the table indicating that uppercase letters denote statistically significant differences among samples within the same column, while lowercase letters indicate statistically significant differences among samples within the same column as well. Including this explanation will help ensure clarity in interpreting the statistical comparisons(Table 2, Line 277-278). Please apply these corrections consistently across all tables and figures that present differences among samples for various parameters. Ensure that the necessary explanations and notations are included wherever relevant to maintain clarity and uniformity throughout the manuscript.

# Please check and correct the title. As 3.1. Determination of Water Absorption Capacity (WAC) (Wwater Aabsorption Acapacity (WAC).

Good luck with corrections...

Author Response

Dear Reviewer 1,

Please find attached the responses to your second review.

Reviewer 2 Report

Comments and Suggestions for Authors

The authors have incorporated the comments suitably and the manuscript now looks fine and is ready for publication. 

Author Response

Dear Reviewer 2,

Thank you for considering our responses satisfactory and recognizing that the work is ready for publication.